# Ultrasonic Treatment of Corn Starch to Improve the Freeze-Thaw Resistance of Frozen Model Dough and Its Application in Steamed Buns

**DOI:** 10.3390/foods12101962

**Published:** 2023-05-12

**Authors:** Rui Han, Jiaqi Lin, Jingyao Hou, Xiuying Xu, Saruna Bao, Chaoyue Wei, Jiayue Xing, Yuzhu Wu, Jingsheng Liu

**Affiliations:** National Engineering Research Center for Wheat and Corn Deep Processing, College of Food Science and Engineering, Jilin Agricultural University, Changchun 130118, China; hanrui@mails.jlau.edu.cn (R.H.); linjiaqi@mails.jlau.edu.cn (J.L.); houjingyao@mails.jlau.edu.cn (J.H.); baosaruna@mails.jlau.edu.cn (S.B.); weichaoyue@mails.jlau.edu.cn (C.W.); xingjiayue@mails.jlau.edu.cn (J.X.); wo_shiwuyuzhu@jlau.edu.cn (Y.W.); liujingsheng@jlau.edu.cn (J.L.)

**Keywords:** ultrasonic, corn starch, frozen model dough, freeze-thaw resistance, buns

## Abstract

Modification of corn starch using ultrasonic waves to improve its freeze-thaw resistance in frozen model doughs and buns. Analysis was performed by rheometry, low-field-intensity nuclear magnetic resonance imaging, Fourier infrared spectroscopy, and scanning electron microscopy. The results showed that the addition of ultrasonically modified corn starch reduced the migration of water molecules inside the model dough, weakened the decrease of elastic modulus, and enhanced the creep recovery effect; the decrease in α-helical and β-fold content in the model dough was reduced, the destruction of internal network structure was decreased, the exposed starch granules were reduced, and the internal interaction of the dough was enhanced; the texture of the buns became softer and the moisture content increased. In conclusion, ultrasound as a physical modification means can significantly improve the freeze-thaw properties of corn starch, providing new ideas for the development and quality improvement of corn-starch-based instant frozen pasta products.

## 1. Introduction

With the accelerated pace of people’s life, frozen products are increasingly favored by consumers. During the transportation and sales process, the quality of frozen dough often deteriorates due to temperature changes, so improving the quality characteristics of frozen dough has become a hot research topic. The internal components of dough in frozen products are complex, and starch and protein, as the main components, play a vital role in the quality of dough [1]. Some studies have used starch and gluten protein to reconstitute flour model doughs to improve freeze-thaw stability in order to mitigate the quality deterioration of frozen doughs. Zhou et al. [2] studied the effect of different particle sizes of potato starch on the quality changes of model dough during freeze-thawing, and the results showed that the model dough with small particle size starch granules had a denser network structure and reduced mobility of internal water molecules. Han et al. [3] reported that type A wheat starch model doughs were more resistant to freezing and thawing than type B wheat starch model doughs under freezing conditions. Therefore, the starch fraction has a greater influence on the freeze-thaw resistance properties of frozen doughs.

As the first major grain crop in China, corn has been widely used in the food industry because of its rich nutritional value. Corn starch is the main component of corn, with a content of about 70%. Due to the defects of starch such as poor freeze-thaw stability and easy aging, it seriously limits its application. To improve freeze-thaw stability, enhance food quality, and expand application areas, raw materials are often treated using physical, chemical, or biotechnological means. Chemical treatment tends to introduce new groups and is not desirable from a food safety point of view [4,5]. The addition of enzymes in biological means is not easy to control, and the degree of damage to the starch structure is greater [6,7]. Therefore, physical modification is still a green, safe and efficient method.

Ultrasonic is a kind of sound wave with a frequency higher than 20 KHz, which is used to modify starch granules through the force of cavitation effect and mechanical effect, and is widely used in food and pharmaceutical fields [8]. The study pointed out that ultrasound treatment caused changes in the particle size and particle structure of starch. Chan et al. [9] studied that ultrasonic treatment of potato starch, corn starch, mung bean starch, and sago starch resulted in amylose chain breaks and increased straight-chain amylose content. Hu et al. [10] used ultrasound to treat the dough and found that the microstructural damage of the dough was reduced and the freezing time was shortened when the ultrasound power was 360 W. Zhang et al. [11] studied the changes of ultrasonic treatment on the moisture distribution of frozen wheat dough and found that the relaxation time was shortened, the freezable water content was reduced, and the internal moisture status of the dough was improved. However, studies related to the properties of frozen dough composed of ultrasonically treated corn starch have not been reported.

This study was conducted to investigate the improvement mechanism of ultrasonically modified maize starch on the freeze-thaw properties of frozen model doughs. The ultrasonically modified corn starch-gluten protein reconstituted model dough was used to investigate the improvement effect of ultrasonically modified corn starch on the anti-freeze-thaw properties of the model dough and its application in steamed buns. By analyzing the rheological properties, moisture distribution, and microstructure of the ultrasonically modified corn starch model dough during the freeze-thaw process, the reasons for the changes in the quality characteristics of its steamed buns were elucidated, and the mechanism of the changes in the anti-freeze-thaw characteristics of the ultrasonically modified corn starch model dough was revealed. It aims to provide a theoretical basis for the development and quality control of convenient quick-frozen grain-starch-based products.

## 2. Materials and Methods

### 2.1. Materials

Corn starch was provided by Shanghai Ruiyong Biotechnology Co., Ltd. of China (corn starch, analytical purity > 98%). Gluten protein was provided by Henan Relent Biotechnology Co., Ltd. of China (protein content > 90%). Dried yeast was obtained from Angel Yeast Co Ltd., Hubei, China. All reagents are analytical grade.

### 2.2. Preparation of Ultrasonically Modified Corn Starch and Frozen Model Dough

#### 2.2.1. Preparation of Ultrasonically Modified Corn Starch

A solution of corn starch with a mass fraction of 6% (*w*/*w*) was prepared and mixed well by magnetic stirring. Ultrasonic power of 120 W and ultrasonic time of 20 min (4 s of probe action and 1 s of interval) were used to treat and obtain ultrasonically modified corn starch.

#### 2.2.2. Preparation of Frozen Model Dough

The ratio of starch to protein in commercially available flour was measured to be 85:15 (*w*/*w*), and the ultrasonically modified corn starch was reconstituted with gluten protein in this ratio. The ratio of flour to water was 2:1. The fully kneaded model dough was kept stable at room temperature for 20 min and evenly divided into several small doughs. The dough was frozen at −20 °C for 22 h and thawed at 30 °C for 2 h as a freeze-thaw Cycle (FT). The freeze-thaw cycle lasted 5 times. The ultrasonically modified corn starch model doughs were denoted as UCS-G, FTU1-G, FTU2-G, FTU3-G, FTU4-G, and FTU5-G. The natural corn starch model doughs were denoted as CS-G, FTC1-G, FTC2-G, FTC3-G, FTC4-G, and FTC5-G.

### 2.3. Determination of Freezable Water Content of Model Dough

The freezable water content of frozen model dough was determined using differential scanning calorimetry (Q2000, TA Instrument, New Castle, DE, USA) [12]. The dough sample (10 mg) was weighed in a crucible. The sample was cooled from 20 °C to −20 °C at a rate of 5 °C/min and held at −20 °C for 10 min and then heated to 20 °C at a rate of 5 °C/min. Calculated by the following equation:(1)$$\mathrm{FW}=\frac{\Delta \mathrm{Hm}}{\Delta \mathrm{Hi}\times \mathrm{Wc}}\;\times 100\%$$
where ΔHm is the enthalpy of the heat absorption peak of the dissolution curve; ΔHi is the known enthalpy of ice melting (334 J/g); and Wc is the moisture content of the model dough.

### 2.4. Determination of Moisture Distribution in Model Dough

The moisture distribution state of frozen model dough was analyzed using low-field-strength MRI (MESOMR23-040V-I, Shanghai Newmark Electronic Technology Co., Shanghai, China) [13]. The transverse relaxation time T_2_ of the sample was determined using a multi-pulse echo sequence (CPMG) and the pulse parameters were as follows: TR = 2000 ms, TE = 0.250 ms, NECH = 10,000, SW = 100 KHz, PRG = 2, and NS = 16.

### 2.5. Determination of Texture Properties of Model Dough

Analysis of the qualitative properties of frozen model dough was performed using a physical property analyzer (TA-XT Plus, TA Instrument, USA). A physical property analyzer with a P/0.5 cylindrical probe was used to analyze the textural properties of the model dough. The velocity was 3 mm/s before, 1 mm/s during, and 5 mm/s after the test.

### 2.6. Measurement of Rheological Properties of Model Dough

#### 2.6.1. Dynamic Rheology Measurement

Determination of dynamic rheological properties of frozen model dough was carried out using a rheometer (Physica MCR-302, Anton Paar, Graz, Austria) [14]. About 2 g of sample was placed on a 25 mm diameter plate rheometer with a gap of 1 mm, a temperature of 25 °C, a scanning strain of 1%, and a measurement angular frequency range of 0.01~100 rad/s. To investigate the variation of elastic modulus (G′), viscous modulus (G″), and loss angle tangent (tanδ) with angular frequency, the elastic modulus (G′) and viscous modulus (G″) are fitted using the power law equation:G′ = k′ω^n′^
(2)
 G″ = k″ω^n″^
(3)
where G′ is the elastic modulus, G″ is the viscous modulus, ω is the angular frequency, and k′, k″, n′ and n″ are constants.

#### 2.6.2. Creep Recovery Measurement

Creep recovery characteristics of frozen model doughs were analyzed using a rheometer (MCR-302, Anton Paar, Austria) [15]. The creep test was performed by applying a constant stress of 50 Pa to the thawed dough samples at a temperature of 25 °C for 300 s. The dough was then allowed to recover for 300 s. The creep recovery curves were fitted to the Burgers model. The creep recovery curve data obtained were fitted to the Burgers model.

### 2.7. Determination of Free Sulfhydryl Content and Disulfide Bond Content of Model Doughs

The method was adopted from Zhao et al. [16] and Fan et al. [17] with modifications. Briefly, Tris-Gly buffer 1 was prepared as 10.4 g Tris, 6.9 g glycine and 1.2 g EDTA dissolved in 1000 mL deionized water, and pH adjusted to 8.0. Tris-Gly buffer 2 was configured as buffer 1 with 8 M urea. A 50 mg sample of lyophilized model dough was weighed and dissolved in 5 mL of Tris-Gly-SDS solution (90 mL of buffer 1 and 10 mL of 2.5% mass fraction sodium dodecyl sulfate solution) and vortexed for 10 min. The sample was vortexed and centrifuged for 10 min at 16,000 rpm/min in a water bath at 25 °C for 1 h. The supernatant was mixed thoroughly with 40 μL of Ellman’s solution (4 mg DTNB and 1 mL of buffer 1), and the supernatant was measured at 412 nm for 30 min in a water bath under dark conditions to determine the free sulfhydryl (SH) content. After centrifugation, 4 mL of supernatant was taken, 1% β-mercaptoethanol was added, vortexed for 10 min, then 40 μL Ellman’s solution (4 mg DTNB and 1 mL buffer 2) was added and mixed thoroughly, and the absorbance of the solution was measured at 412 nm for 30 min in a water bath at 25 °C under dark conditions to determine the total sulfhydryl group (SHT) content.
(4)$$\mathrm{SH}=\;\frac{75.53\times A412\times D}{C}$$
(5)$$\;S-S=\frac{\mathrm{SHT}-\mathrm{SH}}{2}$$
where A412 is the absorbance at 412 nm, D is the dilution factor, C is the protein concentration, and S-S is the disulfide bond content.

### 2.8. Determination of FTIR Spectra of Model Doughs

The secondary structure of the frozen model dough was analyzed using Fourier infrared spectroscopy (VERTEX 70, Bruker Saarbrücken Company, Saarbrücken, Germany). The model dough was freeze-dried under vacuum, mixed with KBr at a ratio of 1:100 (*w*/*w*), and scanned 64 times from 4000 to 400 cm^−1^ at a resolution of 4 cm^−1^. Deconvolution of the amide I bands was performed to analyze the secondary structure of the model dough during the freeze-thaw cycle.

### 2.9. Determination of the Microscopic Morphology of the Model Dough

The microstructure of the frozen model dough was characterized using scanning electron microscopy (Phenom company, Rotterdam, The Netherlands). The samples were fixed for gold spraying, and the apparent morphology was observed at a voltage of 3.0 KV with a magnification of ×500.

### 2.10. Model Dough for Steamed Buns

The method was adopted from Bai et al. [14] and Fan et al. [17] with modifications. First, 1% dry yeast was added to the model dough, let the kneaded fermented dough stand at room temperature for 10 min, freeze it at −20 °C for 22 h, thaw it at 30 °C for 2 h as a freeze-thaw cycle, and repeat 5 times. The thawed model dough was put into a constant temperature and humidity chamber (temperature 35 °C, humidity 85%), woken up for 40 min, and steamed for 25 min. Ultrasonically modified corn starch buns were noted as UCS, FTU1, FTU2, FTU3, FTU4, and FTU5, and natural corn starch buns were noted as CS, FTC1, FTC2, FTC3, FTC4, and FTC5, respectively.

### 2.11. Apparent Characteristics of Steamed Buns

#### 2.11.1. The Color and Specific Volume of the Bun

Steamed buns were photographed and measured using a Sightline Benchtop tabletop vision system.

#### 2.11.2. Textural Properties of Buns

Analysis of the textural properties of steamed buns using a physical property analyzer (TA-XT Plus, TA Instrument, USA). The bun samples were analyzed using a physical property analyzer with a P36R probe. The speed was 3 mm/s before the test, 1 mm/s during the test, and 3 mm/s after the test.

### 2.12. Determination of Moisture Content in Steamed Buns

The steamed buns were allowed to stand at room temperature for 30 min, and a sample of about 2 g was taken from the central part, dried to constant weight at 105 °C, and weighed. The calculation formula is as follows:(6)$$\mathrm{water}\;\mathrm{content}=\frac{2-m}{2}\;\times \;100\%\;$$
where *m* is the mass of drying to constant weight.

### 2.13. Statistical Analysis

Data are the means of 3 replicate trials and are expressed as x ± s. Different letters represent significant differences (*p* < 0.05) and were plotted using SPSS 25 for ANOVA and Origin 2020 for plotting.

## 3. Results

### 3.1. Analysis of Freezable Water Content of Model Dough

The water molecules in the dough exist in many ways, and the main component of freezable water is free water. The formation of ice crystals is one of the main factors causing deterioration in the quality of frozen dough, and the amount of ice crystals depends on the amount of freezable water [18]. Therefore, the lower the freezable water content, the less the network structure of the frozen dough is damaged during the freeze-thaw cycle. As shown in Table 1, with the increase in freeze-thaw times, the content of freezable water gradually increases, indicating that the size and quantity of ice crystals increase, and the physical damage gradually increases. Ding et al. [19] also showed that the freeze-thaw process increases the size of the crystals inside the dough. The freezable water content in the ultrasonically modified corn starch model dough was lower than that of the natural corn starch model dough at the same number of freeze-thaws, and the freezable water content of FTU4-G was 2.49% lower than that of FTC4-G (*p* < 0.05). The cavitation effect of ultrasonic treatment exposes more free radicals in corn starch and increases the interaction with water molecules and protein molecules during the freeze-thaw process. This interaction binds the moisture inside the dough, inhibits the conversion of non-freezable water, and reduces the freezable water content [20].

### 3.2. Analysis of Model Dough Moisture Distribution Results

The quality of frozen pasta products mainly depends on how the internal moisture is distributed during freezing and thawing, and low field strength NMR technology can visually analyze the state of moisture presence. Figure 1 represents the transverse relaxation time profiles of ultrasonically modified corn starch model doughs at different numbers of freeze-thaw cycles and the content of water molecules. The transverse relaxation time spectrum shows three peaks, in which water bound to protein, starch, and other macromolecular components in the dough is strongly bound water, whose content is represented by A21; water indirectly bound to starch, protein, or strongly bound water components by hydrogen bonding is weakly bound water, whose content is represented by A22; water not bound to any component and in a free state is free water, whose content is represented by A23 [21].

From Figure 1A–E, it can be seen that the T_2_ curve of the ultrasonically modified corn starch model dough shifted to the left during the freeze-thaw process and the water mobility became weaker. As can be seen from Table 1, the proportion of free water in the dough increased after freeze-thaw cycles compared to the fresh dough, and the free water of FTU1-G was 30.51% lower than that of FTC1-G (*p* < 0.05), which is consistent with the results of freezable water content. The reason may be that the ice crystals produced by freezing and thawing weaken the binding force of proteins, starches, and other substances in the dough to water, thus affecting the distribution of water inside the dough. Several studies have also shown that the freeze-thaw process increases the release and redistribution of water molecules [12,22]. At the same number of freeze-thaw cycles, the A21 content and A22 content of the ultrasonically modified corn starch model dough were higher than that of the natural corn starch model dough, and the A23 content was lower than that of the natural corn starch model dough. Zhang et al. [23] showed that ultrasonic treatment of the dough increased the bound water content and shortened the relaxation time. The reason may be that ultrasonic changes the structure of corn starch. During dough kneading, the contact area between corn starch and gluten protein is increased, and a closer network structure is formed by cross-linking, which reduces the water loss of dough. Thus, the ultrasonically modified corn starch improved the water distribution state of the frozen dough by redistributing the water molecules.

### 3.3. Analysis of the Results of the Qualitative Properties of the Model Dough

The hardness, springiness, cohesiveness, and reversibility of the model dough were changed during the freeze-thaw cycle and characterized by a physical property meter. As shown in Table 2, the hardness of the ultrasonically modified corn starch model dough gradually increased and the springiness, cohesiveness, and repulsive properties all tended to decrease during the freeze-thaw cycle compared with the fresh dough. The reason may be that during the freezing-thawing process, the water mobility of the dough increases and is lost in the form of free water, destroying the network structure of the dough and causing the hardening phenomenon. The springiness, cohesiveness, and adhesive properties of frozen non-fermented dough decrease when frozen and thawed, while the hardness increases [14,24]. At the same number of freeze-thaw cycles, the hardness values of the ultrasonically modified corn starch model dough were significantly lower than those of the natural corn starch model dough, and the springiness, cohesiveness, and recovery properties were higher than those of the natural corn starch model dough. Among them, the hardness value of FTU4-G decreased by 7.25% (*p* < 0.05) and the springiness value of FTU3 increased by 14.93% (*p* < 0.05) compared with FTC4-G. The water in the dough is mainly composed of strong and weak bound water, and this phenomenon may be caused by the addition of ultrasonically modified corn starch, which enhances the hydration of each component in the model dough through hydrogen bonding. Zhang et al. [23] also showed that hardness and springiness are related to the interactions between macromolecular substances in the dough network structure.

### 3.4. Analysis of Rheological Properties of Model Dough Results

#### 3.4.1. Analysis of Dynamic Rheological Results

The dough rheological property is a major factor affecting the product quality; the elastic quality of dough is expressed by the elastic modulus G′, the viscous quality of dough is expressed by the viscous modulus G″, and the loss angle tangent value tanδ is the ratio of viscous modulus to elastic modulus, which represents the proportion of polymer and reflects the interaction between the components in the dough [25]. That is, a smaller tanδ value represents a greater degree of aggregation.

As can be seen from Figure 2, Appendix A, and Appendix A, the G′ and G″ values of the model dough after freeze-thaw cycles are lower than those of the fresh dough, and the tanδ values are higher than those of the fresh dough, and the tanδ values are less than 1. During the freeze-thaw process, both G′ and G″ values showed a gradually decreasing trend, tanδ values showed a gradually increasing trend, and the polymer proportion decreased. The study reported that frozen dough showed similar results with increasing freezing time [15]. The reason may be that the fine ice crystals in the model dough are gradually transformed into large ice crystals during repeated temperature changes, resulting in water analysis, and the cross-linking effect is reduced, causing the viscoelastic characteristics of the model dough to weaken and present a solid nature [6,14]. Under the same freezing-thawing frequency, the G’ value and G” value of ultrasonically modified corn starch model dough were higher than those of natural corn starch model dough, and the tanδ value was lower than that of natural corn starch model dough, and the polymer proportion and viscoelasticity of the dough increased. In the model dough system formed by ultrasonic modification of corn starch and gluten protein, the skeleton of the network structure is gluten protein, and the ultrasonic modification of corn starch is used as filler to redistribute the water molecules inside the dough during the repeated freeze-thaw process. In our previous research results, it was confirmed that during the freeze-thaw cycle, the seepage degree of amylose content of ultrasonically modified corn starch was reduced, which inhibited recrystallization, so that the polymer ratio between ultrasonically modified corn starch and gluten protein was high, which weakened the damage to the dough network structure [11,26].

The viscoelasticity of the ultrasonically modified corn starch frozen model dough was further analyzed by obtaining fitting parameters through power law equations. The k′ and k″ values are positively correlated with the strength of the model dough, and the n′ and n″ values represent the molecular interaction forces in the model dough [27]. As shown in Table 3, the k′ and k″ values of the model dough showed a decreasing trend with the increase in the number of freeze-thaw cycles, and their values were lower than those of the fresh dough, reflecting the structural weakening and strength reduction of the model dough. Moreover, the values of n′ and n″ gradually increase with the number of freeze-thaws, indicating that the forces within the model dough weaken and the spatial network structure is unstable [14]. Under the same freeze-thaw frequency, the k value of ultrasonic modified corn starch model dough is higher than that of natural corn starch model dough, and the n value is lower than that of natural corn starch model dough, indicating that the dough composed of ultrasonic modified corn starch can reduce the physical damage caused by ice crystals, weaken the internal water migration, and increase the internal structural stability during the freeze-thaw cycle. It has good rheological properties.

#### 3.4.2. Creep Recovery Characteristics Results Analysis

Creep recovery characteristics reflect the trend of rheological changes in model dough under constant pressure and the recovery trend of model dough deformation when constant pressure is removed. As shown in Figure 2G–I, the creep recovery curve of the model dough during freeze-thawing was lower than that of the fresh dough, and the curve of the ultrasonically modified corn starch model dough after the third freeze-thawing was significantly higher than that of the natural corn starch model dough. The fitted data are shown in Table 3, the maximum creep flexibility Jmax is related to the stiffness of the model dough, and the larger the value of Jmax, the less stiff the dough [28]. The zero shear viscosity η_0_ represents the fluidity of the model dough when the shear stress is withdrawn. The Jmax values of the ultrasonically modified corn starch model doughs were higher than those of natural corn starch, and the Jmax values of FTU4-G increased by 12.61% (*p* < 0.05) compared to FTC4-G, indicating a reduced degree of dough hardening. It was noted that under the same freezing conditions, the Jmax value decreased as the freezing time increased [15]. The ratio of Je/Jmax represents the relative elasticity, and the ratio of Jv/Jmax represents the relative viscosity, where the larger the value of Je/Jmax, the stronger the recovery ability of the model dough after deformation, and the smaller the physical damage caused by freezing-thawing [29]. During repeated freezing-thawing, the Je/Jmax values of the ultrasonically modified corn starch model dough were higher than those of the natural corn starch model dough, and the trend of Jv/Jmax values was reversed, reflecting that the ultrasonically modified corn starch model dough had better recovery ability and less deformation when resisting temperature changes.

### 3.5. Analysis of the Results of Free Sulfhydryl Content and Disulfide Bond Content of Model Doughs

The quality of dough depends on the tight network structure formed between starch and gluten protein, and the disulfide bond and free sulfhydryl group can be converted to each other, which is conducive to maintaining the stability of protein molecular structure and directly reflects the stability of dough [17]. Figure 3 shows the relationship between the changes in free sulfhydryl content and disulfide bond content of ultrasonically modified corn starch model dough under different numbers of freeze-thaw cycles. Compared with the fresh dough, the free sulfhydryl content of the model dough increased significantly and the disulfide bond content decreased significantly after freeze-thawing, indicating that freeze-thawing damaged the structure of the dough. As the number of freeze-thaw cycles increased, the free sulfhydryl content of the natural and ultrasonically modified corn starch model doughs showed an increasing trend, while the disulfide bond content showed the opposite. The recrystallization and water migration generated during repeated freeze-thaw cycles were responsible for the disulfide bond rupture, which increased the degree of disruption of the network structure of the dough. Rombouts et al. [30] showed that the oxidation of free sulfhydryl groups is the key to the formation of disulfide bonds and is closely related to the hydroxyl groups in starch. At the same number of freeze-thaw cycles, the free sulfhydryl content of the ultrasonically modified corn starch model dough was lower than that of the natural corn starch model dough, and the disulfide bond content was higher than that of the natural corn starch model dough, probably because the ultrasonically modified corn starch maintained the dynamic balance between sulfhydryl and disulfide bonds between gluten proteins through intermolecular or intramolecular forces, weakening the breakage of disulfide bonds, and increasing the stability of the dough during the freezing process [11]. Some studies also showed that the free sulfhydryl content of the dough gradually increased with the gradual decrease in freezing temperature at the same freezing time. Among them, the free sulfhydryl content of FTU3-G decreased by 16.64% (*p* < 0.5) and the disulfide bond content of FTU3-G increased by 21.59% (*p* < 0.05) compared to FTC3-G. Thus, the ultrasonically modified corn starch resulted in the inhibition of the conversion between sulfhydryl groups and disulfide bonds in the model dough, attenuating the extent of internal structural damage.

### 3.6. Analysis of FTIR Results of Model Dough

Figure 4A–E shows the infrared profiles of the model dough at different numbers of freeze-thaw cycles in the wavelength range of 4000~500 cm^−1^. A stretching vibration peak can be clearly seen at 3350 cm^−1^, which is related to the formation of hydrogen bonds [31]. The positions and shapes of the absorption peaks of the model doughs in the plots were approximately the same, and no new groups appeared. However, it was obvious that the Fourier infrared spectrum of the model dough shifted to the left significantly after the freeze-thaw cycle, and the ultrasonically modified corn starch model dough shifted to the left more than the natural corn starch model dough, indicating that the amount of hydrogen bond formation was greater than the natural corn starch. The hydroxyl group of starch binds to the carbonyl group of protein and amino acid residues by hydrogen bonding [2]. Therefore, the addition of ultrasonically modified corn starch can enhance the cross-linking strength inside the model dough and make the structure more compact.

Table 4 presents the changes in the protein secondary structure content of the model dough during the freeze-thaw cycle. Compared with the fresh dough, the α-helical and β-turned corner contents inside the model dough were significantly reduced after freeze-thawing, while the β-folded and irregularly curled contents changed in the opposite trend. This may be due to pressure differences caused by water migration as the dough freezes and thaws and physical damage caused by recrystallization breaks the disulfide or hydrogen bonds that hold the dough together. Wang et al. [32] demonstrated that freeze-thaw cycles disrupt the molecular structure, and the reduction in hydrogen bonds is responsible for the aggregation and stretching of secondary structures. At the same number of freeze-thaws, the content of α-helix and β-fold in the ultrasonically modified corn starch model dough was higher than that of the natural corn starch model dough, with the α-helix content of FTU2-G increasing by 2.82% (*p* < 0.05) compared to FTC2-G and the β-folding content increased by 4.48% (*p* < 0.05) compared to FTC5-G. It showed that the hydrogen bonding interaction force in the ultrasonically modified corn starch model dough was enhanced and the viscoelasticity of the dough was increased, which was consistent with the rheological results in this paper and with the results of Zhou et al. [2]. After the second freeze-thaw cycle, the irregular curl content of ultrasonically modified corn starch model dough was lower than that of natural corn starch model dough. Freeze-thaw process exposes hydrophilic or hydrophobic groups to proteins, and the previous test results have proved that ultrasonically modified corn starch can reduce the rearrangement of short amylose by inhibiting the formation of double helix structure during freeze-thaw process. Therefore, ultrasonically modified corn starch can increase water cooperation in the model dough. Some research results also proved that ultrasonic promoted the rearrangement of each component molecule in dough and reduced the damage to network structure [11].

### 3.7. Analysis of Microstructure Results of Model Dough

The microstructures of the model doughs under different numbers of freeze-thaw cycles are shown in Figure 5 and Appendix A. Figure 5A and a shows fresh dough, corn starch and gluten protein show a complete network structure, smooth and flat, and the network structure of UCS-G is more regular than that of CS-G. The microscopic network structure damage of the model dough gradually increased during freeze-thawing, and ice crystal pores and grooves could be clearly observed, and the pores gradually became larger. At the fifth freeze-thaw, the starch granules were exposed more obviously. Freeze-thaw cycles can cause irregularities and breakage of the dough structure [23,33]. Therefore, mechanical damage from ice crystal growth is the main cause of the sparse structure of the model dough. However, ultrasonically modified corn starch model doughs have a reduced degree of network structure breakage during the freeze-thaw process, weakening the water mobility and reducing the physical damage produced during the freeze-thaw process. Ultrasound can rough up the surface of corn starch granules and the starch granule size is related to the dough network structure [8,34]. Therefore, the reason affecting the change in the network structure of the ultrasonically modified corn starch model dough may be that the rough starch granules are more tightly bound to the gluten proteins during kneading, effectively weakening the extent to which water molecules penetrate the gluten network structure during freezing and thawing, ensuring the integrity of the structural skeleton and reducing the precipitation of starch granules.

### 3.8. Apparent Characteristics of Buns

#### 3.8.1. The Color of the Bun

Table 5 presents the bun color parameters, where L* represents the degree of black and white, a* represents the red and green values, b* represents the yellow and blue values, and △E represents the combined color of the samples [35]. Compared with fresh steamed bread, L* value decreased and a* value and b* value increased after freeze-thaw, indicating that the whiteness of steamed bread decreased and the surface was yellowish brown. The reason may be that the color of steamed bread is dark, and the ability to reflect light is reduced. The whiteness of steamed buns made from ultrasonically treated corn starch was significantly higher compared to natural corn starch buns at the same number of freeze-thaws. The uneven network structure of dough during freezing-thawing process, the degree of internal moisture migration, and the color change of the dough surface will cause the change in the color of the steamed bun surface [36]. The L* value of FTU4 was 2.92% higher than that of FTC4 (*p* ˂ 0.05), and the ultrasonically modified corn starch buns had a better appearance.

#### 3.8.2. Specific Volume of the Bun

Specific volume is the ratio of volume to mass, which represents the fluffiness, tissue state, and quality of the buns. As shown in Table 6, the specific volume of ultrasonically modified corn starch buns was higher than that of natural corn starch buns when they were not frozen and thawed, indicating that the steamed buns made of corn starch were softer and fluffier after ultrasonic treatment. As the number of freeze-thaw cycles increased, the specific volume of the buns gradually decreased. The reasons for this result may be the physical damage to the gluten network structure of the frozen dough from ice crystals, yeast inactivation, and quality deterioration during the freeze-thaw process [37]. The specific volume of steamed buns made from ultrasonically modified corn starch was significantly higher than that of natural corn starch buns under the same number of freeze-thawing, and the specific volume of FTU1 increased by 9.3% compared with that of FTC1 (*p* ˂ 0.05), indicating that the ultrasonically modified corn starch dough could attenuate the damage produced by the freeze-thawing process, and the dough had strong air-holding property and the quality of buns was improved [38].

#### 3.8.3. Textural Properties of Buns

The texture characteristics directly characterize the quality of the buns, the harder the hardness the worse the taste, and the lower the elasticity the harder the texture [39]. As shown in Table 6, the hardness and chewiness of the steamed buns made from frozen and thawed dough increased significantly and the elasticity decreased significantly compared with the fresh buns as the number of freeze-thawing times increased. The reason for this may be the reassociation of the straight-chain starch during freezing, resulting in faster aging and water analysis, and then increased hardness and cracking of the skin when steamed. Some studies have shown that the hardness and chewiness of steamed buns are positively correlated with the moisture content [21,40]. The texture of steamed buns made by adding ultrasonically treated corn starch to form dough was significantly improved at the same number of freeze–thaw cycles. Among them, the hardness of FTU3 decreased by 28.48% (*p* ˂ 0.05) and the elasticity increased by 9.47% (*p* ˂ 0.05) compared with FTC3. When the frozen dough was steamed, the water gradually evaporated, causing the hardening of the bun crust.

### 3.9. Water Content of Buns

As can be seen from Table 6, the moisture content inside the buns decreased significantly during the freezing and thawing process, which may be due to the conversion of non-freezable water to freezable water in the frozen dough, resulting in a decrease in the internal water-holding capacity of the dough, leading to an increase in the ice crystal content and a decrease in the internal moisture content of the buns. The occurrence of this phenomenon was also confirmed by Vasafi et al. [41]. As can be seen from Figure 6, the moisture content of the natural corn starch buns was significantly lower than that of the ultrasonically modified corn starch buns at the same freeze-thaw times, probably because the ultrasonically treated corn starch produced free radicals and was embedded in the gluten protein network structure in the form of granules, while the gluten protein was hydrophilic and the protein was interlinked with the starch, increasing the internal water-holding capacity. Starch and protein tend to bind to non-freezable water and retard the water migration rate [22,42]. The moisture content of FTU4 was increased by 8.39% (*p* ˂ 0.05) compared to FTC4, with less moisture loss in the buns, resulting in increased internal moisture content.

## 4. Conclusions

In this study, the dough was reconstituted by ultrasonically modified corn starch and gluten protein to investigate the improvement effect of ultrasonically modified corn starch on the anti-freeze-thaw properties of frozen model dough and to study the improvement effect of its quality characteristics of steamed buns. Compared with the natural corn starch model dough, the transverse relaxation time curve of the ultrasonically modified corn starch model dough shifted to the left; the freezable water content of FTU4-G was 2.49% lower than that of FTC4-G (*p* < 0.05); the mobility of water molecules was reduced; the polymer ratio increased and the creep recovery process had a higher Jmax value; the hardness value of FTU4-G was 7.25% lower than that of FTC4-G (*p* < 0.05); less dough hardening; less conversion between free sulfhydryl groups and disulfide bonds; less precipitated starch particles; more regular network structure; enhanced internal interactions; and more stable structure. Compared with natural corn starch buns, the specific volume of ultrasonically modified corn starch buns increased, the hardness of FTU3 decreased by 28.48% compared with FTC3 (*p* < 0.05), the texture became softer and the internal water-holding capacity increased. In conclusion, the addition of ultrasonically modified corn starch improved the rheological properties of frozen model doughs, inhibited the migration of water molecules, reduced the degree of microstructural damage, significantly improved the freeze-thaw resistance of frozen doughs, expanded the application areas of corn starch, and provided a theoretical basis for the development of starch-based instant frozen pasta products.

## Figures and Tables

**Figure 1 foods-12-01962-f001:**
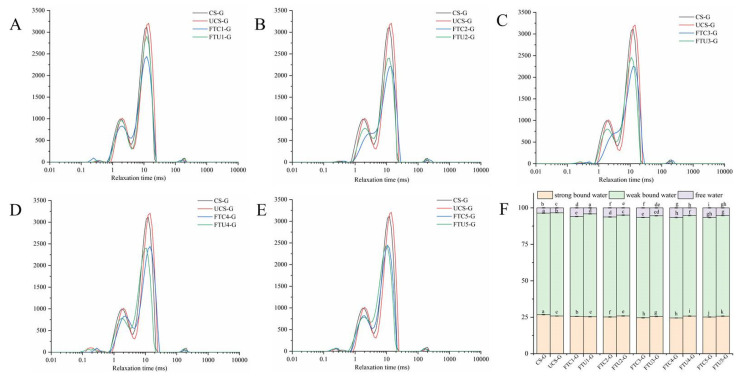
Transverse relaxation time curves and peak area ratios of ultrasonically modified corn starch model doughs at different numbers of freeze-thaw cycles. (**A**–**E**) are the transverse relaxation time curves and (**F**) is the peak area ratio of each component water molecule.

**Figure 2 foods-12-01962-f002:**
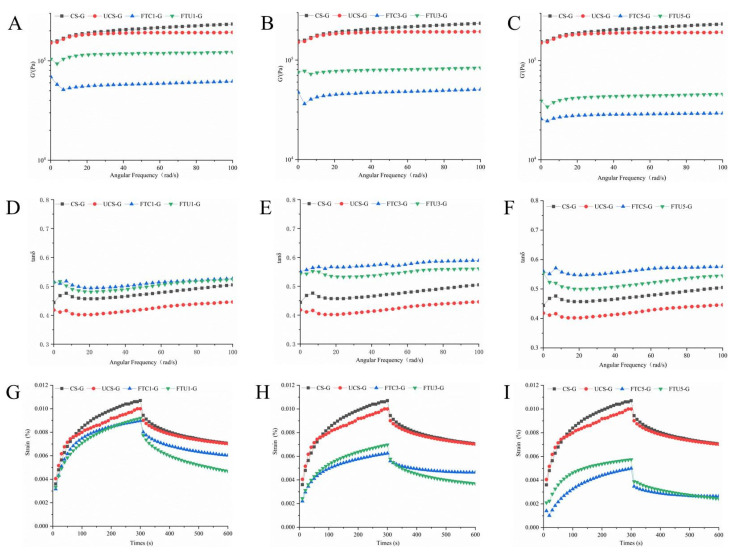
Modulus of elasticity, loss angle tangent, and creep recovery curves of ultrasonically modified corn starch model dough at 0, 1, 3, and 5 freeze-thaw cycles. (**A**–**C**) are the elastic modulus curves, (**D**–**F**) are the attrition angle tangent curves, and (**G**–**I**) are the creep recovery curves.

**Figure 3 foods-12-01962-f003:**
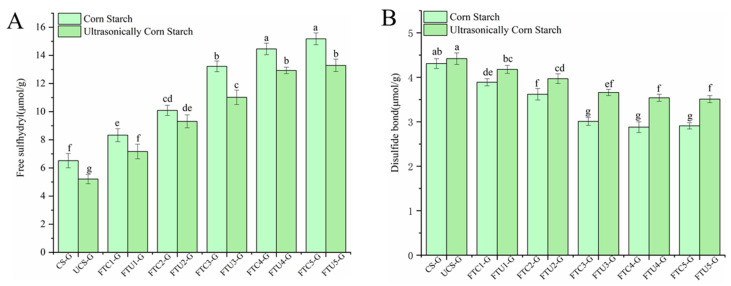
Free sulfhydryl content and disulfide bond content of ultrasonically modified corn starch model dough at different numbers of freeze-thaw cycles. (**A**) is free sulfhydryl content, and (**B**) is disulfide bond content.

**Figure 4 foods-12-01962-f004:**
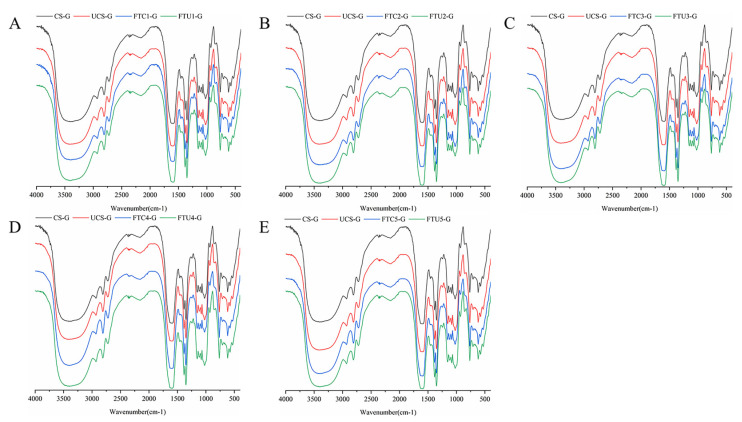
FTIR profiles and secondary structure content of ultrasonically modified corn starch model doughs at different numbers of freeze–thaw cycles. (**A**–**E**) are FTIR profiles.

**Figure 5 foods-12-01962-f005:**
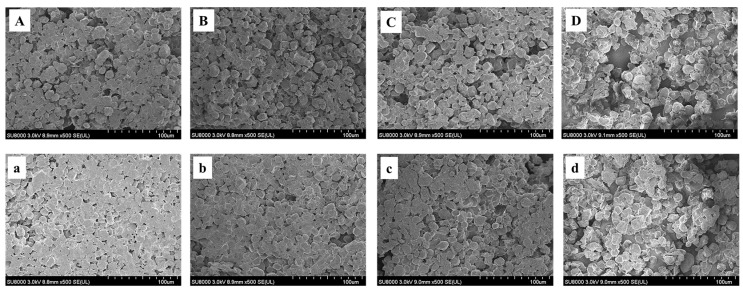
Scanning electron micrograph of ultrasonically modified corn starch model dough at 0, 1, 3, and 5 freeze-thaw cycles. (**A**–**D**) are natural corn starch model dough, and (**a**–**d**) are ultrasonically modified corn starch model dough.

**Figure 6 foods-12-01962-f006:**
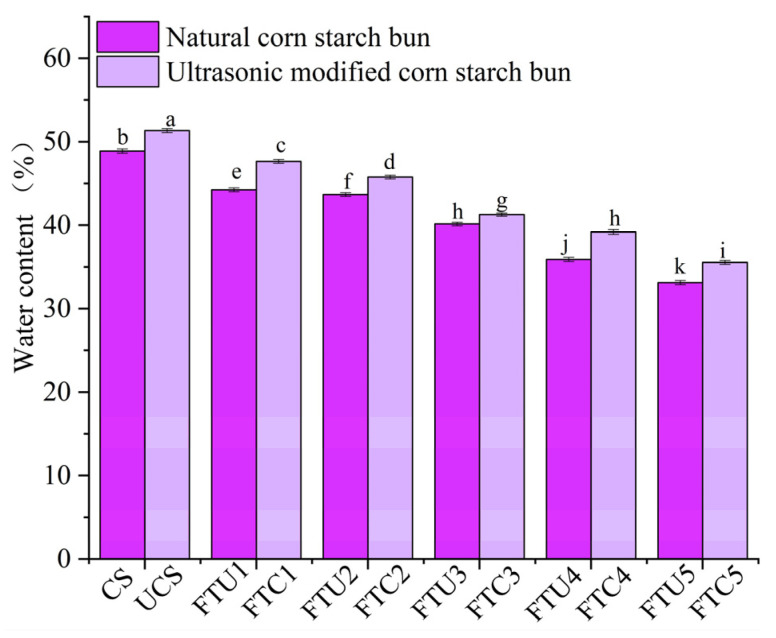
The moisture content of ultrasonically modified corn starch buns under different numbers of freeze-thaw cycles.

**Table 1 foods-12-01962-t001:** The freezable water content and the water molecular content of each component of the ultrasonically modified corn starch model dough at different numbers of freeze-thaw cycles.

	Freezable Water Content (%)	A21 (%)	A22 (%)	A23 (%)
CS-G	31.12 ± 0.56 ^a^	26.749 ± 0.25 ^e^	69.654 ± 0.25 ^f^	3.597 ± 0.28 ^b^
UCS-G	33.01 ± 0.51 ^b^	25.836 ± 0.18 ^f^	70.777 ± 0.16 ^h^	3.387 ± 0.36 ^a^
FTC1-G	39.34 ± 0.45 ^d^	25.619 ± 0.18 ^d^	68.456 ± 0.29 ^b^	5.925 ± 0.26 ^g^
FTU1-G	38.15 ± 0.46 ^c^	25.345 ± 0.26 ^c^	70.548 ± 0.25 ^h^	4.107 ± 0.30 ^c^
FTC2-G	43.78 ± 0.58 ^ef^	25.187 ± 0.30 ^b^	68.568 ± 0.34 ^b^	6.245 ± 0.23 ^h^
FTU2-G	42.61 ± 0.38 ^e^	25.896 ± 0.25 ^f^	69.105 ± 0.39 ^g^	4.999 ± 0.27 ^d^
FTC3-G	47.56 ± 0.39 ^b^	24.658 ± 0.35 ^a^	68.781 ± 0.21 ^c^	6.561 ± 0.31 ^i^
FTU3-G	45.62 ± 0.39 ^g^	25.569 ± 0.34 ^d^	69.008 ± 0.26 ^f^	5.423 ± 0.29 ^f^
FTC4-G	48.29 ± 0.43 ^i^	24.514 ± 0.16 ^a^	68.884 ± 0.36 ^d^	6.602 ± 0.34 ^i^
FTU4-G	45.77 ± 0.41 ^g^	25.793 ± 0.33 ^e^	68.993 ± 0.34 ^e^	5.214 ± 0.18 ^e^
FTC5-G	49.15 ± 0.50 ^j^	25.161 ± 0.22 ^b^	68.241 ± 0.33 ^a^	6.598 ± 0.30 ^i^
FTU5-G	46.94 ± 0.46 ^gh^	25.721 ± 0.38 ^e^	69.101 ± 0.33 ^f^	5.178 ± 0.20 ^e^

Different letters represent significant differences (*p* < 0.05).

**Table 2 foods-12-01962-t002:** Textural properties of ultrasonically modified corn starch model doughs under different numbers of freeze-thaw cycles.

	Hardness(g)	Springiness	Cohesiveness	Resilience
CS-G	1573.22 ± 4.21 ^b^	0.458 ± 0.11 ^g^	0.266 ± 0.01 ^f^	0.075 ± 0.01 ^e^
UCS-G	1233.70 ± 3.75 ^a^	0.342 ± 0.09 ^c^	0.294 ± 0.01 ^g^	0.073 ± 0.01 ^d^
FTC1-G	1705.42 ± 3.33 ^d^	0.403 ± 0.04 ^f^	0.307 ± 0.02 ^h^	0.089 ± 0.01 ^f^
FTU1-G	1543.07 ± 3.56 ^b^	0.382 ± 0.06 ^d^	0.254 ± 0.01 ^e^	0.085 ± 0.02 ^ef^
FTC2-G	1778.43 ± 4.00 ^d^	0.379 ± 0.03 ^d^	0.269 ± 0.01 ^f^	0.073 ± 0.01 ^d^
FTU2-G	1676.62 ± 4.15 ^c^	0.386 ± 0.04 ^e^	0.299 ± 0.01 ^g^	0.077 ± 0.01 ^e^
FTC3-G	1913.41 ± 3.88 ^f^	0.319 ± 0.01 ^a^	0.217 ± 0.02 ^c^	0.067 ± 0.01 ^c^
FTU3-G	1653.09 ± 3.75 ^c^	0.375 ± 0.03 ^d^	0.241 ± 0.03 ^e^	0.073 ± 0.02 ^d^
FTC4-G	1956.35 ± 4.25 ^f^	0.326 ± 0.05 ^b^	0.191 ± 0.01 ^a^	0.061 ± 0.02 ^a^
FTU4-G	1814.49 ± 2.88 ^e^	0.377 ± 0.01 ^d^	0.245 ± 0.02 ^d^	0.064 ± 0.01 ^b^
FTC5-G	2084.02 ± 4.50 ^g^	0.309 ± 0.03 ^a^	0.201 ± 0.01 ^b^	0.061 ± 0.01 ^a^
FTU5-G	1813.63 ± 3.90 ^f^	0.324 ± 0.02 ^b^	0.216 ± 0.01 ^c^	0.065 ± 0.01 ^bc^

Different letters represent significant differences (*p* < 0.05).

**Table 3 foods-12-01962-t003:** Power-law equation fitting parameters and creep recovery fitting parameters for ultrasonically modified corn starch model doughs under different numbers of freeze-thaw cycles.

	k′	n′	k″	n″	Creep Phase		Recovery Phase	
					J_max_ (10^−4^Pa^−1^)	η_0_ (10^−4^Pa^−1^)	Je/J_max_ (%)	Jv/J_max_ (%)
CS-G	2.85 ± 0.08 ^a^	0.14 ± 0.02 ^g^	1.90 ± 0.11 ^d^	0.12 ± 0.01 ^b^	35.27 ± 0.25 ^a^	1.62 ± 0.08 ^h^	47.26 ± 0.33 ^a^	52.73 ± 0.36 ^h^
UCS-G	2.74 ± 0.10 ^d^	0.13 ± 0.05 ^f^	1.74 ± 0.12 ^b^	0.13 ± 0.05 ^c^	33.15 ± 0.27 ^c^	1.53 ± 0.12 ^fg^	45.14 ± 0.21 ^b^	54.85 ± 0.31 ^g^
FTC1-G	2.56 ± 0.11 ^f^	0.15 ± 0.06 ^h^	1.78 ± 0.20 ^a^	0.16 ± 0.06 ^a^	32.18 ± 0.21 ^d^	1.71 ± 0.11 ^gh^	43.35 ± 0.38 ^c^	56.64 ± 0.22 ^f^
FTU1-G	2.63 ± 0.15 ^ab^	0.15 ± 0.03 ^e^	1.83 ± 0.16 ^e^	0.15 ± 0.04 ^fg^	33.89 ± 0.18 ^b^	1.65 ± 0.13 ^fg^	44.76 ± 0.25 ^b^	55.23 ± 0.29 ^g^
FTC2-G	2.51 ± 0.21 ^c^	0.17 ± 0.03 ^g^	1.77 ± 0.23 ^g^	0.18 ± 0.03 ^e^	30.64 ± 0.30 ^e^	1.84 ± 0.07 ^def^	41.46 ± 0.22 ^de^	58.53 ± 0.15 ^e^
FTU2-G	2.56 ± 0.19 ^e^	0.16 ± 0.04 ^f^	1.74 ± 0.18 ^h^	0.16 ± 0.10 ^gh^	30.97 ± 0.33 ^e^	1.76 ± 0.06 ^efg^	43.15 ± 0.26 ^c^	56.84 ± 0.27 ^f^
FTC3-G	2.21 ± 0.16 ^f^	0.19 ± 0.06 ^e^	1.55 ± 0.22 ^f^	0.19 ± 0.11 ^a^	25.41 ± 0.29 ^g^	1.92 ± 0.10 ^bcd^	40.57 ± 0.37 ^e^	59.42 ± 0.22 ^d^
FTU3-G	2.22 ± 0.15 ^h^	0.17 ± 0.09 ^c^	1.62 ± 0.14 ^d^	0.18 ± 0.13 ^h^	28.73 ± 0.45 ^f^	1.87 ± 0.14 ^cde^	41.35 ± 0.31 ^d^	58.64 ± 0.19 ^e^
FTC4-G	2.16 ± 0.14 ^g^	0.20 ± 0.10 ^a^	1.54 ± 0.19 ^g^	0.21 ± 0.07 ^g^	23.49 ± 0.13 ^h^	2.05 ± 0.09 ^ab^	37.99 ± 0.25 ^g^	62.00 ± 0.35 ^b^
FTU4-G	2.13 ± 0.22 ^ab^	0.19 ± 0.08 ^b^	1.57 ± 0.15 ^h^	0.18 ± 0.08 ^a^	26.88 ± 0.22 ^g^	1.98 ± 0.07 ^bc^	39.54 ± 0.33 ^f^	60.45 ± 0.41 ^c^
FTC5-G	1.88 ± 0.13 ^e^	0.23 ± 0.07 ^d^	1.23 ± 0.16 ^f^	0.24 ± 0.04 ^c^	21.87 ± 0.19 ^j^	2.16 ± 0.11 ^a^	36.13 ± 0.36 ^h^	63.86 ± 0.33 ^a^
FTU5-G	1.95 ± 0.17 ^cd^	0.21 ± 0.06 ^c^	1.24 ± 0.23 ^c^	0.22 ± 0.06 ^d^	24.26 ± 0.16 ^i^	2.07 ± 0.08 ^ab^	38.31 ± 0.28 ^g^	61.68 ± 0.39 ^b^

Different letters represent significant differences (*p* < 0.05).

**Table 4 foods-12-01962-t004:** Secondary structure content parameters of ultrasonically modified corn starch model dough under different numbers of freeze-thaw cycles.

	α-Helix (%)	β-Sheet (%)	β-Turn (%)	Random Coil (%)
CS-G	21.65 ± 0.21 ^k^	28.32 ± 0.21 ^b^	30.84 ± 0.23 ^k^	19.19 ± 0.16 ^a^
UCS-G	20.49 ± 0.18 ^h^	28.09 ± 0.14 ^a^	30.34 ± 0.16 ^j^	21.08 ± 0.18 ^d^
FTC1-G	20.87 ± 0.16 ^j^	30.49 ± 0.15 ^e^	28.73 ± 0.18 ^h^	19.91 ± 0.15 ^b^
FTU1-G	20.85 ± 0.23 ^i^	29.47 ± 0.22 ^c^	29.68 ± 0.16 ^i^	20.00 ± 0.21 ^c^
FTC2-G	20.11 ± 0.25 ^g^	30.02 ± 0.19 ^d^	27.85 ± 0.19 ^e^	22.02 ± 0.19 ^g^
FTU2-G	20.39 ± 0.15 ^h^	30.21 ± 0.16 ^d^	28.43 ± 0.14 ^g^	20.97 ± 0.14 ^d^
FTC3-G	19.24 ± 0.16 ^c^	30.87 ± 0.17 ^f^	26.59 ± 0.20 ^b^	23.30 ± 0.13 ^i^
FTU3-G	19.80 ± 0.23 ^f^	30.90 ± 0.13 ^g^	28.11 ± 0.12 ^f^	21.19 ± 0.18 ^e^
FTC4-G	19.61 ± 0.22 ^d^	31.18 ± 0.20 ^h^	26.78 ± 0.26 ^c^	22.43 ± 0.19 ^h^
FTU4-G	19.68 ± 0.15 ^e^	31.50 ± 0.23 ^i^	27.20 ± 0.22 ^d^	21.62 ± 0.21 ^f^
FTC5-G	19.12 ± 0.14 ^a^	30.44 ± 0.14 ^e^	26.88 ± 0.16 ^c^	23.56 ± 0.22 ^j^
FTU5-G	19.15 ± 0.17 ^b^	31.87 ± 0.21 ^j^	26.30 ± 0.18 ^a^	22.68 ± 0.14 ^h^

Different letters represent significant differences (*p* < 0.05).

**Table 5 foods-12-01962-t005:** Color parameters of ultrasonically modified corn starch buns under different numbers of freeze-thaw cycles.

	L*	a*	b*	△E
CS	77.6 ± 0.05 ^a^	5.3 ± 0.04 ^b^	13.2 ± 0.02 ^b^	30.70 ± 0.05 ^b^
UCS	77.4 ± 0.04 ^a^	5.0 ± 0.02 ^a^	12.9 ± 0.04 ^a^	30.49 ± 0.03 ^a^
FTC1	73.2 ± 0.07 ^c^	6.9 ± 0.01 ^d^	14.8 ± 0.02 ^c^	31.45 ± 0.04 ^c^
FTU1	75.1 ± 0.09 ^b^	6.3 ± 0.03 ^c^	14.2 ± 0.01 ^b^	33.42 ± 0.08 ^d^
FTC2	71.7 ± 0.05 ^e^	7.2 ± 0.03 ^f^	15.4 ± 0.03 ^e^	33.81 ± 0.06 ^d^
FTU2	73.1 ± 0.07 ^c^	7.1 ± 0.03 ^e^	15.1 ± 0.03 ^d^	36.39 ± 0.02 ^f^
FTC3	70.1 ± 0.06 ^f^	7.6 ± 0.02 ^h^	15.9 ± 0.02 ^f^	35.65 ± 0.07 ^e^
FTU3	72.1 ± 0.07 ^d^	7.4 ± 0.04 ^g^	15.4 ± 0.02 ^e^	37.11 ± 0.03 ^g^
FTC4	69.8 ± 0.08 ^g^	7.8 ± 0.03 ^j^	16.2 ± 0.02 ^g^	36.12 ± 0.11 ^f^
FTU4	71.9 ± 0.05 ^e^	7.7 ± 0.02 ^i^	15.8 ± 0.01 ^f^	39.47 ± 0.08 ^i^
FTC5	69.5 ± 0.09 ^g^	8.0 ± 0.03 ^k^	16.5 ± 0.03 ^h^	38.12 ± 0.05 ^h^
FTU5	70.1 ± 0.06 ^f^	7.6 ± 0.04 ^h^	16.1 ± 0.01 ^g^	39.89 ± 0.04 ^i^

Different letters represent significant differences (*p* < 0.05).

**Table 6 foods-12-01962-t006:** Specific volume and textural properties of ultrasonically modified corn starch buns under different numbers of freeze-thaw cycles.

	Specific Volume (cm^3^/g)	Hardness (g)	Springiness	Chewiness
CS	1.64 ± 0.02 ^g^	1956.27 ± 30.15 ^d^	1.09 ± 0.05 ^f^	1124.18 ± 13.41 ^b^
UCS	1.69 ± 0.04 ^g^	1339.87 ± 20.19 ^a^	1.11 ± 0.02 ^g^	910.59 ± 24.15 ^a^
FTC1	1.36 ± 0.01 ^e^	2312.56 ± 25.48 ^e^	1.00 ± 0.06 ^e^	1638.74 ± 18.94 ^d^
FTU1	1.50 ± 0.02 ^f^	1589.12 ± 41.38 ^b^	1.01 ± 0.04 ^e^	1356.71 ± 16.23 ^c^
FTC2	1.25 ± 0.02 ^c^	2580.34 ± 52.13 ^f^	0.91 ± 0.02 ^c^	1920.64 ± 20.13 ^e^
FTU2	1.31 ± 0.03 ^d^	1834.25 ± 31.49 ^c^	0.94 ± 0.04 ^cd^	1709.59 ± 25.47 ^d^
FTC3	1.24 ± 0.03 ^c^	2663.44 ± 8.79 ^f^	0.86 ± 0.03 ^b^	2135.38 ± 11.23 ^f^
FTU3	1.29 ± 0.01 ^d^	1904.81 ± 34.65 ^d^	0.95 ± 0.02 ^d^	1801.48 ± 15.61 ^de^
FTC4	1.18 ± 0.04 ^b^	2966.13 ± 38.94 ^g^	0.87 ± 0.03 ^b^	2514.32 ± 21.45 ^g^
FTU4	1.19 ± 0.02 ^bc^	2470.84 ± 29.16 ^e^	0.93 ± 0.03 ^c^	2261.15 ± 19.78 ^f^
FTC5	1.12 ± 0.01 ^a^	3071.96 ± 33.71 ^h^	0.88 ± 0.04 ^b^	2978.45 ± 12.11 ^h^
FTU5	1.18 ± 0.03 ^b^	2746.74 ± 44.21 ^f^	0.79 ± 0.02 ^a^	2499.38 ± 17.45 ^g^

Different letters represent significant differences (*p* < 0.05).

## Data Availability

Data are contained within the article.

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
