# Peer review of "Ultrasonic Treatment of Corn Starch to Improve the Freeze-Thaw Resistance of Frozen Model Dough and Its Application in Steamed Buns"

_foods, 2023, doi:10.3390/foods12101962_

Round 1

Reviewer 1 Report

The manuscript is written well, and the theme is good, but still, there are many minor changes required to improve the manuscript. Here are a few general changes recommended.

The manuscript is scientifically sound, meets the journal's expectations, but still, there are many minor changes required to improve the manuscript. Here are a few general changes recommended.

1.     The abstract needs to be explained properly; therefore, the abstract requires revisions. 

2.. There are numerous typos. Please check throughout the manuscript and add spacing after the line ends. Please make a minor revision of the manuscript that would reorganize and improve its content. Editing for grammar is required. 

3. The material and methods section must be concise, as methods are presented in long paragraphs. 

4. The conclusion requires revision as the results are poorly explained. Overall, the manuscript quality is sound, scientifically effective, and has novelty, and it can be considered for acceptance after minor revision.      

Minor editing of English language is required

Reviewer 2 Report

This manuscript reported on the effect of ultrasonic treatment of corn starch to improve the freeze-thaw resistance of frozen model dough and its application to steamed buns.

The topic is original and  relevant in the field

The study has address a specific gap in the field
This study shows that ultrasonic treatment affected the  freeze-thaw resistance of frozen model dough and its application to steamed buns.

The methodology is sound.

The conclusions are consistent with the evidence and arguments presented. However, it is quite long.

The quality of English is good. However, some improvement is needed as follows:

line 99:  Change to "Determination of.... was performed using differential scanning calorimetry"

line 100: change to " The dough sample (10 mg) was weighed ..."

Line 109: Change to "Analysis of the moisture distribution state of frozen model dough was carried out using low field strength MRI.."

Line115: change to "Analysis of the qualitative properties of frozen model doughs was performed using a physical property analyzer.."

Line 121: change to "Determination of dynamic rheological properties of frozen model doughs was carried out using a rheometer...

Line 249: please rephrase "... as A whole.."

Author Response

Thank you very much for your recognition of the results of this research paper and your recognition of our research area. All of your English speech questions have been changed. Thank you very much for your comments.

Reviewer 3 Report

1. "pre-test" and "pre-experiment" should not be mentioned in the main text of the manuscript.

2. It's better for the authors to explain why 85:15 (w/w) were used in the preparation of frozen model dough. 

3. Tables and Figures should listed the same as the sequence of the results showed in the manuscript. The sequence of them now will confuse readers. 

4. Figure 1 is not clear enough. It's better to list the relaxation time and content of each water fractions in a Table.

5. In table 1, springiness was used for textural properties of dough samples. However, elasticity was used in the main text. They should be in accordance.

6. Figure 4 and Table 3 were used to show the same data. Only one of them was needed, and datas in Table was suggest by the reviewer.

7. Please check the method and data of water content of steamed buns and explain why the water content of buns decreased after FT treatment. The reviewer is doubt about it.

8. The language should be edited extensively. Especially for the results and discussion. It should be more concise, and some important results should be highlighted.

9. Some citations were not use properly. For example, in Line 304, it is the result of this study. Thus no citation was needed.

10. In the method, critical parameters should be descripted in detail. For example, the moisture content of steamed bun was determined immediately or after 1h. It's important for the understanding of the results and for the repeatment of the experiment.

Extensive editing of English language is needed.

Reviewer 4 Report

General remark

This manuscript reports the effect of ultrasonic-treated corn starch on frozen model dough and the application in steamed bun. This manuscript is well-descript and well-discussed. It could be used as fundamental information for application of ultrasonic treatment on bakery industry using corn starch with improved quality. However, some issues need to be addressed before acceptance as follows;

1.    Literature regarding to steamed bud should be added in the “Introduction” section since it is important food model in application part of this manuscript.

2.    Line 83; what is corn starch milk? How does it relate to this work? If it is an ingredient of the dough, please give more specification of this ingredient.

3.    In 2.2.1, please add more information on how to obtain ultrasonic condition.

4.    In 2.13, please add the statistical method used to determine significant differences between treatments.

5.    Table 1, please revise the superscript for all column with consistency. For example, a should be the highest? Superscript a is the highest value for “Freezable water content” and “Hardness” and not for the rest columns.

6.    Table 2, the comment is same to Table 1.

7.    Table 4, there are some incorrect superscript in this table, please check and revise. For example, superscript of a* for FTC5 and FTU5 is not correct. Superscript of b* for FTC4, FTU4, FTC5 and FTU5 is not correct. Superscript of Specific volume for CS and UCS is not correct etc. Please check the rest.

8.    In color determination, delta E (color difference) should be calculated since it can be used for detectable change by human sense.

9.    Figure 6, superscript is not correct. Please check and revise.  

10. Sensory evaluation should be performed to ensure that ultrasonic-treated corn starch is potentially applied in food matrix.

Reviewer 5 Report

This study investigated the freeze-thaw attributes of corn starch/wheat gluten dough after ultrasonic treatment. The data is sufficient and the results are adequate. I will suggest the acceptance of this work after minor revision.

The English language is acceptable. 

Author Response

Thank you very much for your valuable comments and recognition of our research results. The content and conclusion of the article have been reexamined, thank you very much.